# Consumers’ Preferences for Apple Production Attributes: Results of a Choice Experiment

**DOI:** 10.3390/foods12091917

**Published:** 2023-05-08

**Authors:** Ruopin Qu, Jing Chen, Wenjing Li, Shan Jin, Glyn D. Jones, Lynn J. Frewer

**Affiliations:** 1Institute of Agricultural Economics and Development, Chinese Academy of Agricultural Science, Beijing 100081, China; 2School of Natural and Environmental Sciences, Newcastle University, Newcastle upon Tyne NE1 7RU, UK; 3FERA Sciences Ltd., National Agri-Food Innovation Campus, Sand Hutton YO41 1LZ, UK; 4School of Economics and Management, Huazhong Agricultural University, Wuhan 430070, China

**Keywords:** choice experiment, food traceability, apples, random parameter logit, latent class model, Chinese consumers

## Abstract

Various food safety and environmental problems in China have raised consumer awareness of food safety issues and negative environmental impacts in various supply chains. This research assessed consumer preferences and willingness to pay (WTP) for food safety and ecosystem delivery attributes associated with apples, demonstrated through the application of different traceability systems. Research participants were recruited in Beijing (N = 384) and Shanghai (N = 320). Choice experiment methodology was applied. The data were analyzed using conditional logit, random parameter logit, and latent class models; the results indicated significant consumer preferences for traceability information, including in relation to lower pesticide usage and application of organic fertilizer during primary production. The results also indicated that participants in this research had a significant willingness-to-pay premium for apple products that had production information traceability, had reduced pesticide use, and were grown with organic fertilizers. The models demonstrated heterogeneous preferences among participants such that consumers could be divided into three classes: non-price-sensitive (53.5%), pesticide-sensitive (21.7%), and price-sensitive (24.8%).

## 1. Introduction

The quality of food, environmental pollution, and ecological health are becoming increasing societal priorities. In line with this, consumers are becoming increasingly concerned about the use of chemicals in the production of foods [1]. China represents a country with a large agricultural sector, with small farmers representing the largest group of primary producers [2]. There was evidence that farmers usually relied on their own experience to judge the use of pesticides and fertilizers in agricultural production. They did not strictly enforce pesticide and fertilizer application standards, and often aimed to maximize yields in the short term, ignoring the risks to the quality of agricultural products and the environment, which led to the widespread problem of excessive chemical inputs [3] and reduced pesticide resistance in target species [4] . Theoretically, if government subsidies or market premiums can compensate for revenue lost through reduction in yield resulting from reduced input use, farmers would have an incentive to reduce chemical inputs such as pesticides [5]. A reduction in inputs through more precise assessment of agronomic requirements would result in higher farmer incomes (through reduced purchase of chemicals and higher market process for products. Consumers would also have access to better-quality food products in line with (some) consumer references, and the negative environmental impacts of excessive pesticide use in agriculture would be reduced, representing a multi-benefit initiative [5].

Apple is China’s largest cash fruit, primarily produced by smallholder farmers [6]. A common problem associated with apple production in China is the overuse of fertilizer and pesticides [7]. Despite being the largest apple producer globally in terms of quantity, China has made a limited contribution to the international market due to failure to meet international food safety standards because of excess chemical residues (data from FAO, 2019). A previous study examined Chinese consumer preferences and willingness to pay (WTP) for improved food safety [8,9,10], where the primary research focus was on understanding consumer preferences for certificates and brands [11]. To date, there has been less consideration of consumer preferences regarding production inputs and ecosystem protection. The excessive use of pesticides and chemical fertilizers may impact the quality and safety of agricultural products, with detrimental impacts on the farmland environment from an ecological perspective [12,13]. Agricultural environmental pollution would further threaten the quality and safety of agricultural products [14,15]. A reduction in the negative impacts of apple (and other supply chain) production may be facilitated by consumer demand for “pro-environment” (“Pro-environment agricultural products” refer to products that are produced using sustainable and environmentally friendly farming practices. This can include organic farming, agroforestry, regenerative agriculture, and other methods that prioritize soil health, biodiversity, and conservation of natural resources [16] (Octavia et al., 2018)) products.

Traceability systems for agricultural products are an effective means of communicating to consumers about the use of pesticides in the agricultural production process, as well as other production information. China established the first national information platform for the traceability management of agricultural quality and safety in 2017 (https://zycpzs.mofcom.gov.cn/html/moveSY/index.html (accessed on 10 February 2023)), with apples being one of the first traced food products in the country. Included among the production attributes being traced are types of fertilizer applied (organic or inorganic (Organic fertilizer refers to natural materials that are used to improve soil fertility and plant growth, without the use of synthetic chemicals or additives. Inorganic fertilizer refers to synthetic or human-made fertilizers that are typically formulated to provide specific ratios of nitrogen, phosphorus, and potassium. Inorganic fertilizers are often produced using nonrenewable resources, such as fossil fuels, and can be harmful to the environment if overused or applied improperly [17])), the picking date of the fruit, and other production details (such as the “bagging” (Farmers use plastic or paper bags to cover apples during their growth to help with appearance. Bagging is also used because farmers perceive it will prevent apples from absorbing too much pesticide. However, even with bags, systemic pesticides or fungicides will still be transported to the fruit tissues) of apples) Other factors related to the supply chain, such as transport, distribution, or storage periods, can also be traced, including in relation to production practices related to environmental impacts. If consumers are willing to pay a premium for apples that use fewer pesticides, use organic fertilizers, and were produced without the use of bags, this would promote environmental protection and ecosystems, and farmers would be willing to reduce the use of pesticides on their apples to meet consumer and environmental health requirements and international food export standards. Increased consumer demand could motivate apple growers to choose safer, environmentally friendly apple-growing methods, which can potentially be sold at higher prices [18].

Understanding consumer preferences and willingness to pay for traceability attributes and production input attributes could provide data to support producers’ production and pricing decisions. A choice experiment was applied to investigate consumer preferences for traceability and traceability information of apples in order to reveal Chinese consumer preferences and willingness to pay for production-related attributes of apples.

## 2. Literature Review

### 2.1. Guiding Principles of Willingness to Pay

Willingness to pay is a concept related to consumer behavior studies. According to the theory of planned behavior [19], consumer behavior is mainly influenced by a series of behavioral attitudes, subjective consciousness, and behavioral control. Behavioral attitude represents an individual’s belief in the desirability of behaviors, subjective consciousness is an individual’s perceived opinion of what is essential, and behavioral control is an individual’s sense of control over behavior. For Chinese consumers, their willingness to pay for apple production attributes reflects their attitudes toward fresh fruit inputs.

### 2.2. Factors Influencing Consumer Preference

There are a series of factors that can influence consumer preferences; country of origin or region of origin are considered important attributes associated with perceived food quality [20,21,22,23]. This information notifies the consumer about the geographical region or location of production. It is an “experience attribute”, which has emotional meaning to consumers [24] and may not reflect food safety concerns [21,25,26].

There is evidence that consumers value food traceability within supply chains [27,28,29,30,31], but they may have different preferences for what information is traced [32,33]. Attributes included as traceability preferences are “pesticide/veterinary use”, “production date”, and “fertilizer/feed use” [34].

There is evidence that consumers show WTP more for labeled foods such as organic and “green” or pro-environment foods, which may be more sustainable in terms of production practices (e.g., [35,36,37]). Consumers in both resource-rich and low/middle-income countries (LMICs) have reported positive consumer preferences and willingness to pay for pro-environment and organic foods. At the same time, consumers’ willingness to pay more for pro-environment products may vary by product category, with some evidence that they have higher willingness to pay for traditional food categories that are frequently purchased [38] . According to one study, Chinese consumers were willing to pay 47% and 40% more for pro-environment vegetables and pork, respectively [39]. Therefore, there is evidence that Chinese consumers would be willing to pay more for agricultural products produced using more ecofriendly inputs or other pro-environment characteristics.

### 2.3. Certification, Traceability, and Their Functions in Chinese Agrifood Supply Chain

Certification and traceability are the main attributes used to communicate to the public about food safety [29,40]. Certificates can be provided to authenticate food safety for consumers. There are two types of certificates distinguishing agricultural safety level in China, i.e., green certificate and organic certificate, with the latter being stricter on chemical inputs (From the website of the Ministry of Agriculture of PRC, http://www.scs.moa.gov.cn/zcjd/201006/t20100606_1532928.htm (accessed on 12 February 2023)). The national traceability platform sets basic indicators for agricultural product traceability to promote full traceability, including but not limited to production entity (slaughtering and processing), production base, product type, quantity unit, harvest (slaughtering and processing) time, and quality inspection status, as well as the automatically generated product traceability code. The national traceability platform generates two types of certificates, i.e., agricultural product traceability labels with QR codes and edible agricultural product qualification certificates, for agricultural product producers and operators to choose independently (From the website of the Ministry of Agriculture of PRC, http://www.moa.gov.cn/nybgb/2021/202108/202111/t20211104_6381383.htm (accessed on 12 February 2023)).

Even though different consumer segments exist in consumer WTP for traceability and certificates, consumers are willing to pay more for some certificated meat products, of which WTP for government and “green” certificates are the highest [37,41,42]. For agricultural products, studies have indicated higher WTP for certificated food, where the “organic” certificate was highly valued by Chinese consumers [30,43]. Research into traceability preferences for fresh fruit suggested that consumers tended to prefer traceability systems that include production, processing, and distribution information [9].

### 2.4. Growing Fruits and Their Field Requirements

The degree of specialization in apple production is relatively high, and the horticultural process of apple production is more complex. Moreover, the stickiness of labor input during the complete production season is high. A large amount of labor is required for pollination, flower and fruit thinning, bagging, picking, spraying, fertilization, and pruning [44]. In addition to the need for labor, apple plantations require 4–10 pesticide applications and a proper amount of fertilizer to help the apples achieve certain size and color [44]. In addition to the planting experience, personal and family characteristics of apple growers, factors such as financial support and technical training [45], as well as the environmental impact perception of technology [46], could affect the planting management decisions of apple growers.

## 3. Materials and Methods

### 3.1. Theoretical Framework and Experimental Design

The aims were to assess consumer preferences and marginal willingness to pay (WTP) for apple production inputs in first-tier cities in China. According to consumer demand theory [47] and random utility theory [48], consumers’ purchase utility (level of satisfaction derived from a product or service) is derived not from the purchased good but from the attributes associated with the good. Here, the experimental design was based on five attributes associated with the production of apples. Hypothetical scenarios or situations were created to simulate real-life situations in order to test how people might respond to them. Discrete choice experiments (DCEs) are widely used to study consumer preferences for food [49,50,51,52], DCE is an appropriate method for studying consumer preferences and willingness to pay for food attributes (e.g., [32,53,54,55]) Three scenarios (consistent homogeneous consumer preferences, the existence of heterogeneity in consumer preferences, and the existence of market segments among consumers) were analyzed in order to derive consumer preferences and willingness to pay in different contexts. The methodological approach is summarized in Figure 1.

#### 3.1.1. Attribute Selection

Research into consumer preferences for apple attributes addressed certification, traceability, region of origin, organic production brand, taste, and appearance [56,57,58,59,60]. “Pesticide use” and “fertilizer use” for fresh food products were among consumers’ primary concerns for many foods [34]. Research has proven that pesticide use is one major concern of consumers when purchasing apples [24,28,61]. Of the 500 farmers surveyed, 28.6% indicated that they were concerned about the practice of bagging, due to the high labor cost and plastic pollution related to using bags to cover apples. A pilot study (October 2020) investigated consumers’ preferences for information regarding apple production attributes. A total of 140 people participated in the pilot survey. The results suggested that consumers exhibited strong preferences for traced information regarding pesticide and fertilizer used in production (see Appendix B). According to the literature and the pre-survey results, traceability, the number of pesticide applications annually, whether synthetic or organic fertilizers were applied, and whether bags were used or not were selected as attributes; Table 1 presents the details of the attributes and their levels. Traceability attributes included whether traceability technology was applied to the apple supply chain and, if so, what kind of tracing technology was used.

The attribute levels were selected on the basis of the farmer survey results (2019, Appendix A). After eliminating outliers, the minimum number of annual applications of pesticides was four (2.8% of farmers, the low pesticide group), the maximum number of applications was 10 times (3.4% of farmers, the high pesticide group), and the intermediate number of applications was seven times (32.4% of farmers, average number of annual applications, the medium pesticide group). This was to ensure that the research reflected the levels of input used in primary production practice, For similar reasons, the price levels selected were determined by the market price identified in supermarkets, local grocery stores, fruit stores, and e-commerce platforms. The lowest, medium, and highest prices were chosen to represent the prevailing consumer prices at the time of data collection.

#### 3.1.2. Experimental Design

Considering the attributes and levels selected for apples, a total of 4 × 3 × 2 × 2 × 3 (144) scenarios could be generated if a full factorial design was used. According to Allenby and Rossi [63], respondents tire after being exposed to 20 scenarios, with a detrimental impact on the quality of results. To mitigate these problems, a high-efficiency design using Stata 15.1 was used, generating eight scenarios with two alternatives and one opt-out option to make the design more realistic and to retain higher-quality data [9,55]. The D-efficiency for the experiment design was 84.7%, indicating design adequacy. Choice alternatives were demonstrated using pictures with text descriptions of attribute levels to enable respondents to better understand each scenario [64,65]. An example choice set is shown in Figure 1. The survey also collected sociodemographic information (age, gender, education, and family composition). According to a previous study, sociodemographic factors are significant in influencing consumer preferences [32,66].

Apple bagging can improve fruit smoothness and coloring index, as well as reduce appearance defects such as insect eyes; however, it can also reduce the aroma, soluble solids, and total soluble sugar content, increase organic acid content, and overall fail to improve the internal quality of the fruit [67,68]. The use of bags enhances the apples’ appearance and protects them from being sprayed directly with pesticide, but apple bagging also requires high levels of labor. This information was provided to the respondents before they started the experiment. To reduce the bias of appearance differences between varieties and place of origin, the pictures of the apples used in the experiment were taken by the authors (Chen and Qu) in the same orchard for apples of the same variety (Fuji apples) grown with and without bags (Figure 2).

#### 3.1.3. Data Collection

The survey was conducted in Beijing and Shanghai in October–November 2020. At the end of 2019, the population of Beijing was 21.54 million and that of Shanghai was 24.28 million (CRSY). To ensure the quality of the questionnaire, the minimum number of respondents required for Beijing and Shanghai was 308 and 311 respectively, after calculating the sample size required to be representative of consumers in these two cities (Equation (7)). The survey was conducted by trained data collectors. Respondents were selected at random in Beijing and Shanghai. The investigators, accompanied by the authors (Chen or Qu, at least one author at each investigation), stopped pedestrians at random and asked them if they would like to participate in the experiment. To reduce bias, a “cheap talk” technique was used. That is, before the respondent started the questionnaire the author had a short conversation with them, informing them of the purpose and content of the study, and that completing the questionnaire meant that they agreed to participate in the research. The participants were also aware that the data would be analyzed anonymously. Each respondent who participated in the experiment was given a small gift at the end (worth approximately 5 RMB (5 RMB = 5 CNY)). The survey began by asking sociodemographic questions, followed by questions about traceability preferences and apple attributes. A total of 400 questionnaires were distributed in Beijing and Shanghai; 380 questionnaires were collected in Beijing and 370 in Shanghai. After excluding invalid questionnaires, 704 valid questionnaires were collected, resulting in a response rate of 87%.

### 3.2. Modeling Technique

Consumer preferences were analyzed using a random parameter logit model (RPL model) and a latent class model (LC model). The conditional logit model (CL model) assumes that all consumers have the same preferences, which is usually not the case, while the mixed logit model (ML model) or the RPL model relaxes the restriction on identical preferences, so that the ML model and the RPL model can be more flexible in estimating different consumers’ preferences. In addition, latent class models (LC models) are often combined with the RPL model to explain heterogeneity in consumer preferences, as LC models can classify participants on the basis of their preferences and can explain differences in their sociodemographic characteristics.

It was assumed that consumers would seek the combination of attributes that maximize their utility within their purchasing budget. Utility *U_mn_* is consumer *m*’s choice for the commodity *n*, composed of the utilities associated with the commodity’s attributes *V_mn_* and the stochastic error term that captures the unobservable utility *Ɛ_mn_.* An opt-out option was given to the respondents where they could choose neither of the options provided, i.e., *ASC_m_*. The function for the utilities is expressed as
*U_mn_* = *V_mn_* + *Ɛ_mn_*,
(1)

*V_mn_* = *β_ASCm_ASC_m_* + *β*_1_*X*_1*mn*_ + *β*_2_*X*_2_*_mn_* + *β*_3_*X*_3_*_mn_* + … + *β_a_X_a__mn_*,
where *V_mn_* denotes the total utilities of the observed a (a = 1, 2, 3,… a) attributes from *n*, *β* is the weight for each attribute, and X represents the attributes of the commodity. *ASC* is a constant, acting as a dummy variable (0 if consumers choose either option in the choice set; 1 if neither is chosen). When *Ɛ_mn_* follows a type I extreme value distribution, assuming an independent and identical distribution (*iid*), then all consumers are assumed to have identical preferences for the commodity. A conditional logit model (CL model) can be used in this case to estimate the probability that commodity *n* will be selected:(2)$$\mathrm{Pmn}=\frac{\exp\left(\beta^{\prime }mVmn\right)}{\sum_{c}\beta^{\prime }mVmj}.$$

As opposed to the CL model, which assumes that consumers all have the same preferences, the random parameter logit model (RPL) relaxes the strict requirement for consumer preferences and assumes heterogeneity in consumers’ choices. The latent class model (LC) captures the differences in respondents’ preferences and divides them into groups. RPL and LC models are common approaches in choice analysis [6,69,70].

The consumer parameter *β*_m_ cannot be observed directly but can be expressed by the density function ƒ(*β*_m_). The density function ƒ(*β*_m_), integrated with *β*_m_, can be used to obtain the unconditional probability of consumer *m* choosing commodity *n* in the RPL model:(3)$$\mathrm{Pmn}=ƒ\left(\beta m\right)d\beta m\times \int \frac{\exp\left(\beta^{\prime }mVmn\right)}{\sum_{c}\beta^{\prime }mVmj},$$
where β_m_ is the parameter vector, *V_mn_* is the utility vector, and c is the number of alternatives in a choice set.

When ƒ(*β*_m_) is discrete, the probability of consumer *n* falling into class k choosing *m* is
(4)$$\mathrm{Pmn}=\mathrm{Qnk}\times \sum_{k=1}^{k}\frac{\exp\left(\beta^{\prime }kVmn\right)}{\sum_{c}\beta^{\prime }kVmj},$$
where *β_k_* is class k’s parameter vector, and Q_nk_ is the probability of consumer *n* belonging to class k. Q_nk_ can be estimated by
(5)$$\mathrm{Qnk}=\frac{exp\left(Zn\partial^{\prime }k\right)}{\sum_{r=1}^{R}exp\left(Zn\partial^{\prime }k\right)},$$
where *Zn* is a series of observed values influencing consumer *n* in a certain class, and $\partial^{\prime }k$ denotes the parameter vector of consumer in class *k*.

Assuming that the variables and constants follow normal distribution, and that the utility function is linear, WTP for a particular attribute is the ratio of the attribute variable parameter in the utility function to price variable parameters:

mWTP = −β_any attribute_ ÷ β_price_.
(6)


Dummy variables were employed for the non-price variables; thus, the marginal WTP of an attribute level can be interpreted as the WTP of consumers for attributes compared to the baseline level. The baseline level for traceability was no-trace, and the WTP was compared with blockchain-trace, high-trace, and medium-trace. Similarly, the baselines were medium-pesticide for pesticides, chemical for fertilizer, and with-bag for bagging.

The Scheaffer formula [71,72] was used to determine the required sample size to represent the population:(7)$$n=\frac{N}{1+\left(N-1\right)\times \sigma^{2}}÷(0.8),$$
where N is the population of the city chosen, and σ was chosen to be 6%. The scrap rate of questionnaires was assumed to be 20%; n represents the minimum sample size needed.

## 4. Results and Discussion

### 4.1. Descriptive Statistics

The sample characteristics are described in Table 2. Respondents in this study came from two cities, Beijing and Shanghai, with slightly more respondents from Beijing than from Shanghai, accounting for 55% of the full sample. In terms of gender, the proportion of women was higher than for men, with 60% female respondents, making the sample slightly unbalanced in terms of gender., Young respondents accounted for a relatively high proportion, with respondents aged 25 and below accounting for 41% of the total sample, respondents 25 to 40 years old accounting for 39%, respondents 40 to 55 years old accounting for 14%, respondents 55 to 70 years old accounting for 5%, and respondents over 70 years old accounting for 1%. From the perspective of family income, the respondents with middle income accounted for the highest proportion, while the respondents with a monthly income of 5000–20,000 CNY and 20,001–35,001 CNY accounted for 46% and 20% of the total sample, respectively. Respondents with a monthly income of <5000 CNY accounted for 17%, and respondents with a monthly income of >35,000–50,000 and 50,000 accounted for 9% and 7%, respectively. By the end of 2019, the average disposable income per capita was about 73,800 CNY in Beijing and about 73,600 CNY in Shanghai (Data source: Official website of the China Bureau of Statistics).

The average income of the sample in this study was about 21,000 CNY, in line with the distribution of disposable income per capita in Beijing and Shanghai after necessary expenses were excluded. In terms of education level, 27% of respondents in the sample had completed junior high school or below, 7% had completed senior high school, 39% had a bachelor’s degree or equivalent qualification, and 27% had a postgraduate degree or above. The average education level of the respondents was slightly higher than the Beijing/Shanghai average, mainly due to the relatively greater willingness of those with higher education levels to take such surveys. In terms of household members, 63% of respondents had children at home, 2% had pregnant women at home, 55% lived with their parents, and 29% lived alone.

### 4.2. CL and RPL Model Results

#### 4.2.1. Homogeneity and Heterogeneity Preference

The estimation results for the CL model with fixed preferences and the RPL model with heterogeneous preferences are shown in Table 3. Since the RPL model relaxes the assumption that *Ɛ_mn_* follows a type I extreme value distribution and is independently identically distributed (iid), consumers can have different preferences for each attribute. After testing various distributions and comparing existing similar studies [9,55], normal distribution parameters were specified for the attributes in the RPL model. As shown in Table 3, the coefficient for the ASC option (Chooseno) was negative and significant in both models, implying that the utility of not choosing either option was less than the utility derived from choosing either option. As expected, the coefficient of price was significantly negative in both models, suggesting that higher prices significantly reduce consumers’ utility. In both models, the estimated coefficients on all traceability and production attributes of apples were significant at the 1% level, except for the low pesticide attribute in the RPL model, which was significant at the 5% level, indicating that all these attributes have a significant effect on respondents’ preference for Fuji apples. In addition, the standard deviation of the RPL model indicated significant heterogeneity in respondents’ preferences for all apple production attributes other than fertilizer application, but a more consistent preference for traceability. The generally significant standard deviation coefficients demonstrated some variation in consumer preferences for apple production attributes, confirming the validity of using the RPL model for estimation.

The empirical results from the CL and RPL models showed that, although there were some differences in the results they produced, the level of significance was largely consistent with the positive and negative directions, indicating that consumers’ preferences, although somewhat heterogeneous, had a similar general tendency. The coefficients for all three traceability levels were positive, indicating that consumers prefer apples associated with traceability information. The standard deviations for the three traceability attributes were not significant, implying considerable homogeneity in consumers’ evaluation of traceability. Compared to the baseline (medium pesticide application), consumers strongly opposed high pesticide application and preferred low pesticide application. In terms of standard deviation, consumers had a strong heterogeneous preference for high pesticides and a lesser but still significant heterogeneity for low pesticides. Consumers preferred apples with organic fertilizers used in the production process to chemical fertilizers. However, consumers showed a positive preference for bagged apples, and the difference in preference for bagging attributes was also significant.

#### 4.2.2. Factors Influencing Pesticide Application Preferences

On the basis of the the pilot survey, which revealed that the production input of most concern to consumers was pesticides, further research was conducted on pesticide input preferences by constructing an interaction term between low pesticide options and respondent demographic characteristics to derive characteristics that influence consumers’ choice of low pesticide attributes (Table 3). The results of both the CL model and the RPL model showed that age had a significant positive effect, i.e., older respondents were more likely to choose apples with low pesticide attributes. Gender characteristics were dummy variables, with 1 being set to male and 0 to female, and the results showed that female respondents were more likely to choose apples with low pesticide attributes. The effect of income on low pesticide was not significant, suggesting that apples, as a relatively affordable popular fruit, had a small range between the highest and lowest selling prices; therefore, there was no significant effect of high or low income on apple preference. In the CL model, respondents with children were more likely to choose low-pesticide apples, but this attribute was not significant in the RPL model, nor was the presence of parents or pregnant women in the household. The presence of parents and pregnant women also had no significant effect on the low pesticide attribute. Age, gender, and education were the main factors influencing the preference for low pesticide attributes.

### 4.3. Latent Class Model Results

The number of classes needs to be pre-decided for the LC model. Following Boxall and Adamowicz [73] and Allenby [74], the minimum Akaike information criterion (AIC) and the minimum Bayesian information criterion (BIC) were used to examine the optimal number of classes that would fit the data. In this case (see Appendix C), both AIC and BIC decreased as the number of classes increased. The model was then run with the number of classes set between two and 10 to compare the level of significance of the parameters. According to the significance of parameters and the marginal improvement of AIC and BIC values, the number of classes was determined to be 3.

Table 4 shows the estimated preferences of the three categories of respondents in the model. The results showed that consumers had a 53.5% chance of falling into category one, a 21.7% chance of falling into category two, and a 24.8% chance of falling into category three. Most of the preference parameters in category one were significant, but the price attribute was not; hence, category one was named “price-insensitive”. Respondents in category one had a significant positive preference for traceability, in addition to a positive preference for low pesticide and organic fertiliser applications and a significant aversion to high pesticide applications. These respondents had a significant preference for both traceability attributes and low chemical inputs in production, and the price coefficient was negative but not significant, implying that their utility in purchasing apples did not significantly decrease with increasing price.

The second group of respondents showed some unique preferences. Respondents in the second category placed less importance on traceability attributes than all other groups, with none of the three levels of traceability having a significant effect on preferences. However, they placed great importance on pesticide use, with high pesticide attributes exhibiting a strong negative correlation for preference and low pesticides exhibiting a significant positive correlation. The second category was, therefore, named “pesticide sensitive”. They also showed a clear preference for the use of bagging techniques. In addition, ASC parameters were not significant, suggesting that pesticide reduction and bagging use were more important for these respondents.

Respondents in the third category were price-sensitive, with the price parameter showing a significant negative correlation with utility. Respondents also showed a significant negative preference for the ASC option, suggesting that they considered the utility of not buying apples to be lower than buying any kind of apples; that is, an increase in price significantly reduced utility. Therefore, respondents were classified as “price-sensitive” consumers. Category three respondents also disliked apples produced without bags more than the other two groups and were not greatly interested in traceability information, with only moderate traceability attributes. This suggested a significant positive correlation, indicating that they were only interested in traceability information related to production inputs.

### 4.4. Willingness to Pay

Consumer willingness to pay for each attribute predicted by each model can be seen in Table 5, with each willingness-to-pay value indicating the highest premium consumers were willing to pay for apples with that attribute. According to the RPL model, consumers were willing to pay 1.471 RMB per 500 g more for apples traced using blockchain technology than for apples that did not carry any traceability information. The premium for the blockchain attribute in the CL model was 6.675 RMB, with price-insensitive, pesticide-sensitive, and price-sensitive consumers willing to pay 51.074, 1.842, and 0.542 RMB, respectively, for this attribute. However, except for the CL model, willingness to pay for blockchain traceability was not significant in the other models, where consumers were not sure they would pay for it. In contrast, the premium for apples traced back to production only was higher and more significant than that for blockchain traceability, suggesting that consumers were most willing to pay for access to information on production inputs for apples.

Specifically in terms of production input attributes, consumers generated a negative willingness to pay for apples with high pesticide use frequency compared to apples with moderate pesticide use frequency, while apples with low pesticide use frequency received a significant price premium. This result was in line with expectations that, as consumers’ living standards improved, so did their demand for food safety and environmentally friendly products, a result that provides ample incentive for apple producers to improve their orchard management capabilities, reduce pesticide use frequency, and adopt more environmentally friendly production practices. Similar to the willingness-to-pay results for attributes related to pesticide use, consumers were willing to pay a premium for apples produced with organic fertilizers compared to apples produced with chemical fertilizers. However, while consumers showed a willingness to pay more for environmental attributes in terms of pesticide and fertilizer attributes, they were not willing to pay for no bagging. The reason for consumers’ reluctance to pay a premium for apples grown without bags may be due to the fact that these apples are generally less bright in appearance than those grown with bags. Changing this perception may require improved apple production techniques and more communication about the negative impacts of bagging on the environment.

The insignificance of the price attribute of the price-insensitive consumers led to an insignificant willingness to pay for all attributes in their category. The willingness to pay for this category, although high, was due to the extraordinarily small parameter of price; it remains to be tested in the market whether a corresponding premium can be achieved for each attribute. Pesticide-sensitive consumers showed a tendency to pay for low-pesticide apples, consistent with their preferences. Price-sensitive consumers showed a willingness to pay a premium for traceability in production and for the use of organic fertilizers, showing that production was indeed the attribute that consumers cared most about.

### 4.5. Policy Discussions from Model Outcomes

CL, RPL, and LC models were used to estimate consumer preferences and WTP for traceability and production input attributes of apples in Beijing and Shanghai. A heterogeneity of consumer preferences was found using the RPL and LC model. The results showed significant heterogeneous consumer preferences for apple production and traceability inputs, using the LC model to categorize consumers, where there was a 53.5% probability of a consumer being price-insensitive, 21.7% probability of being pesticide-reduction-oriented, and 24.8% probability of being price-sensitive.

The results showed that consumers had a significant preference for apples associated with traceability system. Among these, the traceability of production attributes attracted the attention of consumers, and consumers were willing to pay a higher premium for obtaining production input information. The research on the traceability system and the traceability content showed [75] that Chinese consumers preferred a food traceability system with detailed information and were willing to pay for it. Moreover, Chinese consumers were most interested in the quality certificate and the detailed information of chemical fertilizers/pesticides used in food production. In addition, education level, self-reported health status, and risk attitude affect the willingness to pay [9]. Education level also had a significant impact on consumer preference. This finding can be applied to market segmentation and marketing strategies for traceable apples, including publicity methods, pricing strategy, and distributing platforms.

The greatest preference expressed by participants was for traced pesticide information. As indicated by previous research [34,62,76], consumers appeared to prefer reduced pesticide usage in production. However, here, respondents valued appearance almost as much as other traits [76,77], as reported for other fresh fruits [78,79,80,81]. For example, consumers may express a higher willingness to pay for food (fresh strawberries) produced using sustainable production methods because they are considered to be pro-environment, particularly in relation to reduced pesticide application [82]. There is market potential for apples produced using less pesticide in China. Sociodemographic variables may represent important factors influencing consumer preference for pesticide application frequency. The results showed that age, education level, and gender significantly affect consumer preference for pesticide attributes. Participants who were older, had higher levels of education, and were female were associated with preferences for reduced pesticide use in apples.

In contrast, respondents indicated a preference for the use of bags in apple production. This might be because consumers lacked knowledge about the pollution caused by the bags or because they were more concerned about food safety issues compared to environmental impacts, as some participants explained their preference for bagged apples on the basis that it would prevent the apples from absorbing chemicals. Even though consumers were concerned about the quality and safety of vegetables and agricultural environmental pollution, there was evidence that they were more concerned about personal health issues compared to environmental pollution [83].

The results of this research provide evidence-based strategic planning for the development of food safety in the apple industry and potentially other production sectors. The production input of each apple and the marginal price of traceability indicated that consumers in first-tier cities are willing to pay a premium for traceability attributes. Although some participants appeared unaware of the environmental impacts of bags, the preferences and willingness to pay for reduced pesticide use suggested that there is a potential market for apples cultivated using environmentally sustainable methods. At present, this has not translated into changed agronomic practices in primary production. Policies can be developed and implemented to encourage farmers to apply more sustainable practices in relation to apple cultivation. This will increase the availability of apples produced with lower levels of pesticide input compared to conventional farming, potentially driven by consumer demand within the apple supply chain. As consumers are willing to pay more for these products, this may increase farmer income, assuming that increased payment costs are passed down to primary producers.

## 5. Concluding Remarks

Some recommendations can be made for regulatory bodies and stakeholders. First, policies should be developed to encourage apple growers and processors to apply traceability systems to apple production, processing, and distribution. The implementation of traceability systems can address information asymmetries, thereby restoring consumer trust in food quality, in addition to having human and environmental health impacts. As there was evidence of consumer demand for authentication of pro-environment production practices, information about production attributes should be traced and made available to supply chain endpoint consumers, as well as different actors in the supply chain.

Increased investment in research focused on pro-environment production technologies that enable reduced chemical inputs such as fertilizers and pesticides can provide the basis for introducing pro-environment practices on the farm, as well as provide the basis of supply in relation to consumer demand.

Agricultural organizations such as cooperatives can be encouraged to provide pro-environment production services or guidance to increase the likelihood of farmers adopting pro-environment production technologies. It may be important for policy specialists and supply chain stakeholders to work with extension services to provide resources for farmers which enable access to new agricultural technologies allowing a reduction in inputs in primary production via education and information provision.

Some limitations of the research can be identified. First, the participant sample had a higher level of education compared to the Chinese population overall. People with higher education may pay more attention to the environmental impacts of food production and the inputs of their food. If further research indicates that the preferences for pro-environment production do not extend to other socio-economic groups, a more targeted strategy for these products might be relevant. The preferences of consumers in first-tier cities in China were assessed. Further research might usefully examine consumer preferences in a range of cities across China or compare consumers living in rural and urban environments. In addition, the survey did not include respondent preferences for apples associated with size, taste, or certification (e.g., quality certification of agricultural products such as organic certificates). Consumers’ preferences also depend on fruit quality traits such as sweetness and size or certificates or place of origin; it is not known how consumers rank the importance of these attributes relative to production attributes.

## Figures and Tables

**Figure 1 foods-12-01917-f001:**
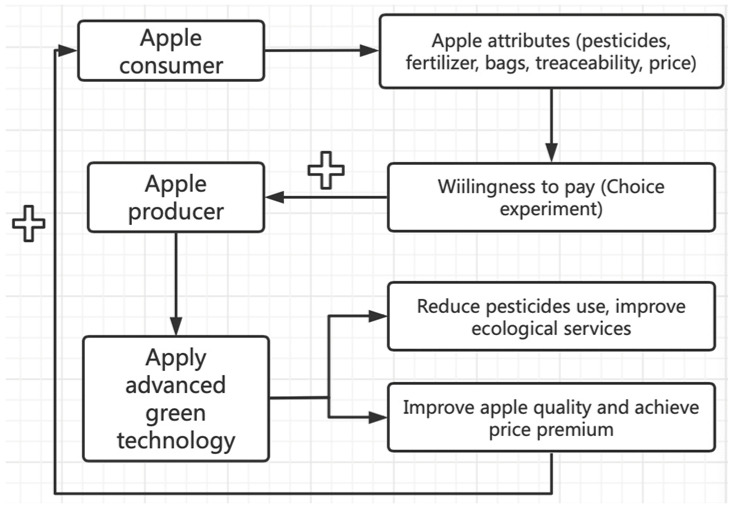
Methodological framework.

**Figure 2 foods-12-01917-f002:**
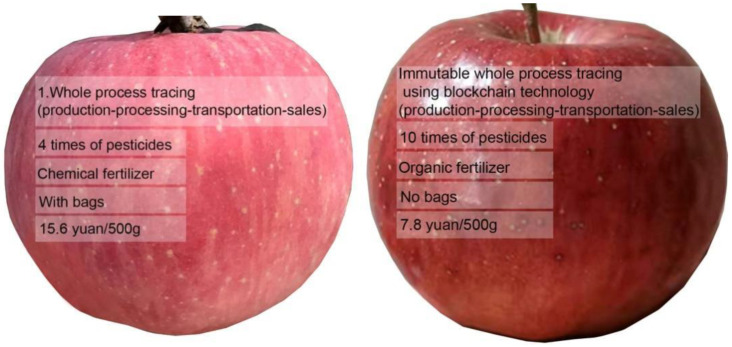
Sample scenario (The photographic illustrations in Figure 2 were taken by the authors during previous surveys of apple farmers in China. To ensure that the appearance of the apples was as unaffected as possible by other factors, red Fuji apples ripening at the same time in the same orchard were used for the pictures, the only difference being whether or not a bag was used during the growing process).

**Table 1 foods-12-01917-t001:** Attributes of apples in the choice experiment.

Attribute	Level	Description
Traceability	4	Unchangeable digital traceability throughout the supply chain (blockchain traceability) Traditional supply chain traceability methods (high traceability) Production traceability (medium traceability) No tracing information (no traceability)
Pesticide use	3	Four applications per year (low pesticide) (Compared with the previous literature that described pesticide usage as “conventional”, “reduced pesticide”, or “limited pesticide” [62], this study used the number of applications per year to demonstrate the real situation of apple production to the respondents. According to the farmers’ survey, they use around 500 L of water to dilute the pesticides for each application. Each application contains a similar amount of pesticides; therefore, higher frequency is equal to more pesticide). Seven applications per year (medium pesticide) Ten applications per year (high pesticide)
Fertilizer	2	Organic fertilizer Chemical fertilizer
Bags	2	With bags Without bags
Price paid by the consumer for apples	3	3.99 CNY/500 g 7.8 CNY/500 g 15.6 CNY/500 g

1 CNY = 0.1508 USD at time of survey.

**Table 2 foods-12-01917-t002:** Descriptive statistics.

Category	Number	Percentage	Category	Number	Percentage
Male	281	40%	City		
Female	423	60%	Beijing	384	55%
Age (years)			Shanghai	320	45%
<25	288	41%	Education		
25–40	273	39%	Junior high or under	188	27%
40–55	101	14%	High school	51	7%
50–70	34	5%	Bachelor or equivalent	273	39%
>70	8	1%	Undergraduate and above	192	27%
Income (CNY)			Family		
<5000	121	17%	Children	447	63%
5000–20,000	327	46%	Pregnant	15	2%
20,001–35,000	138	20%	Parents	388	55%
35,001–50,000	66	9%	Alone	206	29%
>50,000	52	7%			
Total	704	100%	Total	704	100%

**Table 3 foods-12-01917-t003:** CL and RPL model results.

Variables	CL Model	RPL Model	SD
Blockchain traceability	0.874 *** (0.060)	0.755 *** (0.103)	−0.072 (0.097)
High traceability	0.757 *** (0.065)	0.640 *** (0.114)	0.155 (0.127)
Medium traceability	0.895 *** (0.069)	0.716 *** (0.071)	−0.014 (0.223)
High pesticide	−0.352 *** (0.053)	−0.551 *** (0.066)	0.468 *** (0.120)
Low pesticide	1.056 *** (0.166)	0.644 * (0.263)	0.300 * (0.121)
Organic	0.284 *** (0.042)	0.277 *** (0.050)	−0.111 (0.153)
No-bag	−0.477 *** (0.043)	−0.436 *** (0.047)	0.513 *** (0.079)
Price	−0.131 *** (0.005)	−0.165 *** (0.010)	0.206 *** (0.011)
ASC (opt-out)	−3.177 *** (0.181)	−5.192 *** (0.216)	3.403 *** (0.167)
Low pesticide × age	0.296 *** (0.040)	0.236 *** (0.065)	−0.189 ** (0.076)
Low pesticide × gender	−0.265 *** (0.072)	−0.259 * (0.116)	0.133 (0.223)
Low pesticide × income	−0.028 (0.033)	0.015 (0.053)	−0.028 (0.091)
Low pesticide × education	0.072 *** (0.009)	0.040 *** (0.013)	0.037 *** (0.009)
Low pesticide × children	0.148 * (0.075)	0.143 (0.115)	−0.129 (0.198)
Low pesticide × pregnant	0.017 (0.242)	0.044 (0.417)	−0.289 (0.444)
Low pesticide × parent	−0.051 (0.071)	0.125 (0.107)	−0.202 (0.168)
Log likelihood	−8291.7578	−4240.3167	−4240.3167
LR chi^2^	2436.1	2187.64	2187.64
Prob > chi^2^	0.0000	0.0000	0.0000
Total	16,896	16,896	16,896

Standard errors are in parentheses; * *p* < 0.05, ** *p* < 0.01, *** *p* < 0.001.

**Table 4 foods-12-01917-t004:** Latent class model results.

Variables	Category 1 (Price-Insensitive)	Category 2 (Pesticide-Sensitive)	Category 3 (Price-Sensitive)
Blockchain traceability	0.674 *** (0.141)	0.207 (0.185)	0.268 (0.481)
High traceability	0.511 ** (0.154)	0.191 (0.215)	0.783 (0.631)
Medium traceability	0.655 *** (0.075)	0.191 (0.180)	1.239 * (0.571)
High pesticide	−0.502 *** (0.070)	−0.886 *** (0.199)	−0.469 (0.296)
Low pesticide	0.361 *** (0.074)	0.398 ** (0.153)	−0.435 (0.370)
Organic	0.258 *** (0.058)	0.178 (0.144)	0.536 (0.294)
No-bag	−0.320 *** (0.047)	−0.347 ** (0.133)	−0.823 ** (0.286)
Price	−0.013 (0.008)	−0.112 *** (0.019)	−0.494 *** (0.059)
ASC (opt-out)	−3.446 *** (0.163)	−0.184 (0.227)	−6.620 *** (0.701)
Class share	53.5%	21.7%	24.8%

Standard errors are in parentheses; * *p* < 0.05, ** *p* < 0.01, *** *p* < 0.001.

**Table 5 foods-12-01917-t005:** Willingness to pay empirical results (in RMB).

Variable	CL Model	RPL Model	Price-Insensitive	Pesticide-Sensitive	Price-Sensitive
Blockchain traceability	6.675 ***	1.471	51.074	1.842	0.542
High traceability	5.778 ***	−0.057	38.705	1.7	1.584
Medium traceability	6.831 ***	3.383 ***	49.607	1.697	2.506 **
High pesticide	−2.685 ***	−2.987 ***	−38.001	−7.879 ***	−0.95
Low pesticide	8.059 ***	4.520 ***	27.321	3.539 **	−0.88
Organic	2.165 **	0.093	19.529	1.584	1.085 **
No-bag	−3.644 ***	−2.705 ***	−24.253	−3.088 **	−1.665 ***
ASC (opt-out)	−24.249 ***	−37.655 ***	−260.968	−1.64	−13.395 ***

Standard errors are in parentheses; ** *p* < 0.01, *** *p* < 0.001.

## Data Availability

Data is contained within the article.

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
