# Peer review of "Consumers’ Preferences for Apple Production Attributes: Results of a Choice Experiment"

_foods, 2023, doi:10.3390/foods12091917_

Round 1

Reviewer 1 Report

The Authors performed a consumer study on the tracebility of domestically grown apples in China. In the introduction and the Literature review, the cited references give a clear argumentation about the necessity of such a system. From lines 67 we read some details on the national tracing system on apple production. It is mentioned, that the usage of fertilizers are recorded, however it would be very necessary to record the use of pesticides and fungicides also. In this section, the methodology of ‘bagging’ the apples is mentioned. It should be noted, that although the fruit is covered by a bag, the systemic pesticides or fungicides will still be transported to the fruit tissues.

In lines 148-150 the Authors reflect, that there was no study before, where apples were investigated related to fertilizer/pesticide use and consumer attitude. Although the methodology is not the same, but there were several studies in that field, just to mention a few:

Loureiro, M. L., McCluskey, J. J., & Mittelhammer, R. C. (2002). Will consumers pay a premium for ecolabeled apples?. Journal of consumer Affairs36(2), 203-219.

Cerroni, S., Notaro, S., & Shaw, W. D. (2013). How many bad apples are in a bunch? An experimental investigation of perceived pesticide residue risks. Food Policy41, 112-123.

Tsakiridou, E., Mattas, K., & Bazoche, P. (2012). Consumers' response on the labels of fresh fruits and related implications on pesticide use. Food Economics9(1-2), 129-134.

In Section 3.2. the selection of attributes is clear, and supported by a pilot study. This provide a robust study framework, levels are chosen based on practical experiments in that sector.

To footnote 5 I have a comment: even if the amount of water to dilution is the same, it does not necessarily mean, that the persistence and risk of the pesticide or the fungicide is the same.

Figure 2 is a good example, how consumers’ responses were collected. I have an impression, that this choice experiment is very similar to the conjoint analysis protocol.

In line 196 we read “the bagging of apples…did not improve the taste of apples”. Could you please insert a reference here about that sensory test?

In lines 206-207 there is a reference to Equation 6, but I was unable to find it in the manuscript. On the other hand it is not a pre-requisite for a scientific study to provide a fully representative dataset for large populations. But it is necessary for drawing the conclusions with mentioning the limitation of the sample, which takes place between lines 540 and 551. In my opinion 600 respondents cannot truly represent 45 million people, but it is not the goal here. The article is still a good study on this field of science.

Lines 210-211: ‘cheap talk’ means something like informing the respondents on the study goals?

Line 216: 5 RMB is 5 yuan?

Lines 217-218: “Four questionnaires were distributed in Beijing” – I suppose it should be four hundred.

There are a few typos in the text:

Line 69 Instead “tracedare” type „traced are”

Line 132 Instead „from the good purchased” type „ from the purchsed good”

Author Response

Thank you so much for taking your time to review my paper, I deeply appreciate your contribution and suggestions. I have carefully read and taken in your suggestions, here I will explain my responses.

  1. “The Authors performed a consumer study on the tracebility of domestically grown apples in China. In the introduction and the Literature review, the cited references give a clear argumentation about the necessity of such a system. From lines 67 we read some details on the national tracing system on apple production. It is mentioned, that the usage of fertilizers are recorded, however it would be very necessary to record the use of pesticides and fungicides also. In this section, the methodology of ‘bagging’ the apples is mentioned. It should be noted, that although the fruit is covered by a bag, the systemic pesticides or fungicides will still be transported to the fruit tissues.”

Thank you for the information, I have added this in the footnote “Farmers use plastic or paper bags to cover apples during their growth to help with appearance. Bagging is also used because farmers perceive it will prevent apples from absorbing too much pesticide. However, even with bags, systemic pesticides or fungicides will still be transported to the fruit tissues.”

  1. In lines 148-150 the Authors reflect, that there was no study before, where apples were investigated related to fertilizer/pesticide use and consumer attitude. Although the methodology is not the same, but there were several studies in that field, just to mention a few:

Loureiro, M. L., McCluskey, J. J., & Mittelhammer, R. C. (2002). Will consumers pay a premium for eco‐labeled apples?. Journal of consumer Affairs36(2), 203-219.

Cerroni, S., Notaro, S., & Shaw, W. D. (2013). How many bad apples are in a bunch? An experimental investigation of perceived pesticide residue risks. Food Policy41, 112-123.

Tsakiridou, E., Mattas, K., & Bazoche, P. (2012). Consumers' response on the labels of fresh fruits and related implications on pesticide use. Food Economics9(1-2), 129-134.

Thank you for providing these literature, I have added them in line 153: “Pesticide use” and “fertilizer use” for fresh food products were among consumers’ primary concerns for many foods (Liu et al., 2018), research had proved that pesticide use is one major concern of consumers when purchasing apples (Loureiro et al., 2002; Cerroni et al., 2013; Tsakiridou et al., 2012).

  1. In Section 3.2. the selection of attributes is clear, and supported by a pilot study. This provide a robust study framework, levels are chosen based on practical experiments in that sector.

Figure 2 is a good example, how consumers’ responses were collected. I have an impression, that this choice experiment is very similar to the conjoint analysis protocol.

We are honored that you appreciate our selection of attributes, as for collecting consumer responses, choice experiment does have a lot in common with conjoint analysis.

  1. In line 196 we read “the bagging of apples…did not improve the taste of apples”. Could you please insert a reference here about that sensory test?

Thank you for noticing, we have now added some reference here, now this paragraph is “ Apple bagging can improve fruit smoothness, coloring index, and reduce appearance defects such as insect eyes, but it could also reduce the aroma, soluble solids, and total soluble sugar content, increase organic acid content, and overall fail to improve the internal quality of the fruit (J.Wang, 2017; Q.Ma et al., 2021). The use of bags enhances the apples’ appearance and protects them from being sprayed directly with pesticide, but apple bagging also required high levels of labour. This information was provided to the respondents before they started the experiment. To reduce bias of appearance differences between varieties and place of origin, the pictures of the apples used in the experiment were taken by the authors (Chen and Qu) in the same orchard for apples of the same variety (Fuji apples) grown with and without bags .”

  1. In lines 206-207 there is a reference to Equation 6, but I was unable to find it in the manuscript. On the other hand it is not a pre-requisite for a scientific study to provide a fully representative dataset for large populations. But it is necessary for drawing the conclusions with mentioning the limitation of the sample, which takes place between lines 540 and 551. In my opinion 600 respondents cannot truly represent 45 million people, but it is not the goal here. The article is still a good study on this field of science.

We are so sorry as it should have been Equation 7, we were aware that 600 respondents could not truly represent 45 million people so we looked for references for the minimum number of respondents required for such research.

Here is Equation 7 and the explanation for your convenience:

The Scheaffer formula (Cui et al., 2012; Y. Wang et al., 2020) was used to determine the required sample size to represent population:

÷(0.8)      (7)

where N was the population of the city chosen, σ was chosen to be 6%.  It was assumed the scrape rate of questionnaires to be 20%, and n is the minimum sample size needed.

  1. Lines 210-211: ‘cheap talk’ means something like informing the respondents on the study goals?

Yes. “cheap talk” is a technique used to inform the respondents about the study goals and providing them the basic background information of the study so they could better understand the questionnaire.

  1. Line 216: 5 RMB is 5 yuan?

Yes,RMB is yuan. We will add this in the paper to clarify.

  1. Lines 217-218: “Four questionnaires were distributed in Beijing” – I suppose it should be four hundred.

Yes, we have changed it to four hundred, thank you for noticing.

  1. There are a few typos in the text:

Line 69 Instead “tracedare” type „traced are”

Line 132 Instead „from the good purchased” type „ from the purchsed good”

Thank you so much for reading this paper so conscientiously, we have corrected these typo in line 69 and line 132 now.

Reviewer 2 Report

The authors investigated consumers preference and willingness to pay for food safety and ecosystem delivery attrirbutes associated with apples, demonstrated via the application of different traceability systems. The method and findings discussed appears very promising. Some areas to help improve the quality of this piece of work are as follows:
a) The literature review needs further expansion, to help build the backbone to strengthen this case. Please make the literature review to be of three sections, each section with at least two paragraphs, to cover the following:
i) subsection 1 should be captioned: "guiding principles of willingness to pay", here, tell us about the major principles tyhat underscore willingness to pay, how did it emerge, why did it emerge, what are those principles, and how is it perceived in China context
ii) subsection 2 should be captioned: "factors influencing consumer preference", here tell us about these factors, two sentences of each would suffice, narrow it down to end with how it operates in the China contexts
iii) subsection 3 should be captioned: certification, traceability and their functions in Chinese agrifood supply chain, here tell us which certifications are obtained thereof, which traceability approaches are available thereof, how do these two concepts interrelate, how do consumers function within this concepts, which attributes are associated with them, etc...
iv) subsection 4 should be captioned 'growing fruits and their field requirements', here tell us about the needs of apple in the field, fertilizer applications, factors influencing field requirements, etc
The reviewer looks forward to seeing this very important section.
b) Please, 3 and 4, should be merged as one, as Materials and methods, where 3.1. is Theoretical framework and experimental design, and 3.2. Modeling Technique
The reviewer will examine this also
c) Please, sections 5 and 6 should be merged , all under Results and Discussion
make 6.1. should be "Policy discussions from model outcomes, whereas 6.2 should be completely separated, and made  as part of a new section to stand on its own completely 'Concluding remarks'

This is a very brilliant study, and look forward to its revised version

Author Response

Response letter to reviewer 2:

Thank you so much for taking your time to review my paper, I deeply appreciate your contribution and suggestions. I have carefully read and taken in your suggestions, here I will explain my responses.

  1. The literature review needs further expansion, to help build the backbone to strengthen this case. Please make the literature review to be of three sections, each section with at least two paragraphs, to cover the following:
  2. i) subsection 1 should be captioned: "guiding principles of willingness to pay", here, tell us about the major principles that underscore willingness to pay, how did it emerge, why did it emerge, what are those principles, and how is it perceived in China context

  3. ii) subsection 2 should be captioned: "factors influencing consumer preference", here tell us about these factors, two sentences of each would suffice, narrow it down to end with how it operates in the China contexts

iii) subsection 3 should be captioned: certification, traceability and their functions in Chinese agrifood supply chain, here tell us which certifications are obtained thereof, which traceability approaches are available thereof, how do these two concepts interrelate, how do consumers function within this concepts, which attributes are associated with them, etc...

  1. iv) subsection 4 should be captioned 'growing fruits and their field requirements', here tell us about the needs of apple in the field, fertilizer applications, factors influencing field requirements, etc
    The reviewer looks forward to seeing this very important section.

Thank you for your advice, we have rearranged the literature review section:  

  1. Literaturereview

2.1. Guiding principles of willingness to pay

Willingness to pay was a concept related to consumer behavior studies. According to the Theory of Planned Behaviour (Ajzen, 1991), consumer behavior was mainly influenced by series of behavioral attitudes, subjective consciousness, and behavioral control. Behavioral attitudes was an individual’s belief in the desirability of behaviors, subjective consciousness was an individual’s perceived opinion of what’s essential, and behavioral control was an individual’s sense of control over behavior. For Chinese consumers, their willingness to pay for apple production attributes reflected their attitudes towards fresh fruit inputs.

 2.2 Factors influencing consumer preference

There were a series of factors that could influence consumer prefrences, country of origin or region of origin had been considered as an important attribute associated with perceived food quality (Claret et al., 2012; Kendall et al., 2018; Wongprawmas et al., 2015; Zbib et al., 2010). This information informed the consumer about the geographical region or location of production. It is an “experience attribute”, which had emotional meaning to consumers (Ehmke et al., 2008; Tsakiridou et al., 2011), and may not reflect food safety concerns ( Kendall et al., 2018; Kendall, Clark, et al., 2019; Kendall, Kuznesof, et al., 2019).

There was evidence that consumers value food traceability within supply chains (X. Liu et al., 2015; Loureiro & Umberger, 2007; Ubilava & Foster, 2009; Yan et al., 2015; Zhou et al., 2010), but they may have different preferences for what information is traced (Bai et al., 2019; C. Zhang et al., 2014). Attributes included as traceability preferences include “pesticide/veterinary use”, “production date” and “fertilizer/feed use” (C. Liu et al., 2018). 

There was evidence that consumers were WTP more for labelled foods such as organic and “green” or pro-environment foods which may be more sustainable in terms of production practices (e.g Daugbjerg et al., (2014).; Ejvl et al., 2011; Z. Zhang et al., 2013). Consumers in both resource rich,  and low to middle income countries (LMICs) have reported positive consumer preferences and willingness to pay for pro-environment and organic foods. At the same time, consumers' willingness to pay more for pro-environment products may vary by product category, and there was some evidence that they have higher willingness to pay for traditional food categories that were frequently purchased (Krystallis & Chryssohoidis, 2005). In the case of Chinese consumers, Chinese consumers were willing to pay 47% and 40% more for pro-environment vegetables and pork, respectively (Yu et al., 2014). From the former research, there was evidence that Chinese consumers would be willing to pay more for agricultural products that were produced using more eco-friendly inputs or with other pro-environment evidences.

2.3 Certification, traceability and their functions in Chinese agrifood supply chain

Certification and traceability were the main attributes used to communicate to the public about food safety (R. Liu et al., 2013; Ubilava & Foster, 2009). Certificates can be provided to authenticate food safety for consumers. There were two types of certificates distinguishing agricultural safety level in China: green certificate and organic certificate, with organic certificate being more strict on chemical inputs. The national traceability platform sets basic indicators for agricultural product traceability to promote full traceability, including but not limited to: production entity (slaughtering, processing), production base, product type, quantity unit, harvest (slaughtering, processing) time, quality inspection status, and other information, as well as the automatically generated product traceability code. The national traceability platform generates two types of certificates, namely agricultural product traceability labels with QR codes and edible agricultural product qualification certificates, for agricultural product producers and operators to choose independently.

Even though different consumer segments existed in consumer WTP for traceability and certificates , consumers were willing to pay more for some certificated meat products, of which WTP for government and “green” certificates are the highest (Lusk et al., 2018; Wu et al., 2014; Z. Zhang et al., 2013). For agricultural products, studies had indicated higher WTP for certificated food where the “organic” certificate was highly valued by Chinese consumers (Yan et al., 2015; Yin et al., 2015). Research into traceability preferences for fresh fruit had suggested that consumers tended to prefer traceability systems that include production, processing and distribution information (Liu et al., 2020).

2.4 Growing fruits and their field requirements

The degree of specialization in apple production is relatively high, and the horticultural process of apple production is more complex. Moreover, the stickiness of labor input during the complete production season is high. A large amount of labor is required for pollination, bagging, flower and fruit thinning, bagging, picking, spraying, fertilization, and pruning (Ma, 2019). Besides the need for labour, apple plantation required 4 to 10 times of pesticides and proper amount of fertilizer to help the apples achieve certain size and colour (Ma, 2019). In addition to the planting experience, personal and family characteristics of apple growers, factors such as financial support and technical training (Wang and Huo, 2012), and environmental impact perception of technology (Song, 2019) could affect the planting management decisions of apple growers.

  1. b) Please, 3 and 4, should be merged as one, as Materials and methods, where 3.1. is Theoretical framework and experimental design, and 3.2. Modeling Technique
    The reviewer will examine this also

Thank you for your advice, we have changed accordingly.

  1. c) Please, sections 5 and 6 should be merged , all under Results and Discussion
    make 6.1. should be "Policy discussions from model outcomes, whereas 6.2 should be completely separated, and made as part of a new section to stand on its own completely 'Concluding remarks'

    This is a very brilliant study, and look forward to its revised version

Thank you for your advice, we have now rearranged the paper accordingly.

Round 2

Reviewer 2 Report

The authors have made very significant improvements in the work. They have clearly strengthened the introduction, as well as provided clarity in the methodology. The reviewer is satisfied with the current revised version. It is now acceptable for publication.